# Quality of Care in Hospitals and the Use of Mobile-Based Personal Health Record Applications: An Exploratory Study Using National Hospital Evaluation Data

**DOI:** 10.3390/healthcare12111064

**Published:** 2024-05-23

**Authors:** Young-Taek Park, Mi-Joon Lee, Sang Mi Kim

**Affiliations:** 1HIRA Research Institute, Health Insurance Review & Assessment Service (HIRA), Wonju-si 26465, Republic of Korea; pyt0601@hira.or.kr; 2Department of Medical Information, Kongju National University, Gongju-si 32588, Republic of Korea; mijoon1004@kongju.ac.kr; 3Department of AI Health Information Management, Yonsei University, Wonju-si 26493, Republic of Korea

**Keywords:** electronic health record, personal health record, personal medical information, quality of care, mobile health

## Abstract

The use of mobile-based personal health record (m-PHR) applications at the hospital level has been minimally studied. This study aimed to investigate the relationship between m-PHR use and quality of care. A cross-sectional study design was employed, analyzing data from 99 hospitals. Two data sources were utilized: a previous m-PHR investigation conducted from 26 May to 30 June 2022 and a hospital evaluation dataset on quality of care. The use of m-PHR applications was measured by the number of m-PHR application downloads. Three independent variables were assessed: quality of care in the use of antibiotic drugs, injection drugs, and polypharmacy with ≥6 drugs. A generalized linear model was used for the analysis. The hospitals providing high-quality care, as evaluated based on the rate of antibiotic prescription (relative risk [RR], 3.328; 95% confidence interval [CI], 1.840 to 6.020; *p* < 0.001) and polypharmacy (RR, 2.092; 95% CI, 1.027 to 4.261; *p* = 0.042), showed an increased number of m-PHR downloads. Among the hospital covariates, public foundation status and being part of multi-hospital systems were associated with the number of m-PHR downloads (*p* < 0.05). This exploratory study found a positive relationship between quality of care and m-PHR use. Hospitals providing high-quality care may also excel in various activities, including m-PHR application use.

## 1. Introduction

Hospitals are places where a series of activities constantly take place to provide medical services. Most activities aim to improve the satisfaction of patients and their families, including the quality of care, and increase the managerial efficiency of hospital operations. One of the recent trends is that many hospitals have introduced a mobile-based personal health record (m-PHR) system (hereinafter “application”, “app”, or “system”) into their practice settings [1,2,3]. Healthcare institutions, such as hospitals and clinics, provide patients with applications that allow them to view their visit records, schedules, medications, and test results [4,5].

These apps are useful tools for hospitals because they are convenient for patients. By using these apps, patients can see their schedules and the tests they have had in the past [6]. By accessing their records, patients can better understand their health and disease status and actively participate in treatment and prevention activities [7]. For these purposes, hospitals develop or deploy m-PHR apps that patients or their families can download from Google Play and the Apple Store.

A conspicuous phenomenon in the use of m-PHR applications is that the number of downloads varies depending on the hospital. Some hospitals have had 160,000 downloads since their app was launched, while others have had significantly fewer [8]. Why do these variations among hospitals occur? If we can understand the causes of these differences, we can encourage more patients to download the applications. The more downloads, the higher the likelihood that patients will use them. To promote the use of m-PHR systems, we need to identify the factors influencing the number of m-PHR app downloads.

There have been no studies on the causes of this variation or related factors. This study argues that hospitals providing high-quality medical care will have more downloads of m-PHR apps than others by assisting their patients in downloading these applications. The Task–Technology Fit theory may support this argument. The theory postulates that individuals or organizations behave to maximize their performance by aligning their tasks with appropriate technology [9,10,11]. Here, “task” and “technology” could be defined or measured by a hospital’s various activities related to improving the quality of care and “m-PHR application”, respectively. This implies that m-PHR applications are well-suited to activities aimed at enhancing the quality of care, and hospitals interested in quality care are likely to adopt these applications in their operations. Several empirical studies indirectly support this prediction. Hospitals providing good-quality care are more likely to adopt various information technologies [12]. According to previous studies, hospitals that introduced m-PHR systems had better infrastructure in terms of facilities, personnel, and equipment [13]. The opinion also exists that m-PHR systems are a useful tool that can help hospitals provide high-quality medical services [14,15]. M-PHR systems are also known to be effective in chronic disease management [16,17,18,19]. Healthcare institutions with more health information exchanges tend to have a well-composed medical staff [20]. In these respects, hospitals that provide high-quality care may have a higher number of m-PHR downloads than others.

This study aimed to investigate the relationship between the quality of healthcare in hospitals and the use of m-PHRs as measured by the number of m-PHR application downloads. The results are expected to provide useful information for promoting the use of m-PHR apps by identifying the causes or related factors that affect m-PHR downloads and utilizing this information in policy-making.

## 2. Materials and Methods

### 2.1. Study Design

This study used a cross-sectional design, and the unit of analysis was hospitals with >100 beds. The study group consisted of “general hospitals” and “tertiary hospitals”. According to Korea’s national health insurance law, a “general hospital” must have more than 100 beds and at least seven specialized medical departments. Tertiary hospitals share the same characteristics as teaching hospitals. To enhance the generalizability and comparability of the study subjects, hospitals with fewer than 100 beds, classified as “small hospitals”, were excluded. Therefore, the term “hospitals” in this manuscript refers to general hospitals and tertiary hospitals.

Regarding the research design, this study compared the number of downloads of m-PHR apps between hospitals that received a 1st-grade rating in each of the three items used to evaluate the quality of care and hospitals that did not receive a 1st-grade rating in Korea. This study categorized these evaluation results into 1st-grade and non-1st-grade institutions to facilitate an easier interpretation for the reader. The three items evaluated were the antibiotic prescription rate for acute upper respiratory infections, the injection drug use rate, and polypharmacy, defined as prescriptions containing six or more drugs. The concept of polypharmacy, which has been extensively studied academically, refers to the number of prescribed drugs [21]. In Korea, the Health Insurance Review and Assessment Service (HIRA) evaluates prescriptions that include six or more drugs.

The study setting was Korea, which has a national health insurance (NHI) system. All citizens are free to choose a primary clinic-level medical institution and can then visit general hospitals or tertiary hospitals [22]. Because patients can freely choose general hospitals or tertiary hospitals, large hospitals make continuous efforts to retain their patients [23]. Since the m-PHR system offers many functions which provide various conveniences to patients [5,14], it is very likely that hospitals with high-quality medical services will be interested in the introduction and use of m-PHR applications.

### 2.2. Data Sources

Two data sources were used in this study. The first source comprised data from previous studies, which included information on the current status of m-PHR system implementation in hospitals and the number of m-PHR app downloads. The investigation of m-PHR applications in the previous study had been conducted between 26 May and 30 June 2022. Using the hospital enrollment booklet, this study directly examined whether the hospital had a related m-PHR app through the Google Play Store, the Apple Play Store, Naver, and Google search engines. This research method was similar to those used in previous studies [13,24]. A total of 364 general hospitals (including 43 tertiary hospitals) were investigated, of which 101 (27.7%) had introduced the m-PHR system. Two of these institutions were excluded from the analysis because they lacked general characteristics and evaluation results. Finally, 99 hospitals were included in the analysis.

The second source were public data from the HIRA, which included data on the evaluation results of hospitals (hospital evaluation data) and their general characteristics. The HIRA evaluates hospitals using various indices. For this study, three items were selected as they could be comprehensively applied to all patients and are considered representative indicators for evaluating the quality of hospitals’ medical services: the prescription rate of antibiotic use for acute upper respiratory tract diseases, the prescription rate of injection drugs, and polypharmacy evaluated based on the number of medicines prescribed. The HIRA assessed these items, announced the results with grades ranging from 1 to 5, and disclosed the results to the public. The evaluation of these three indicators was based on the number of claims for which a review decision was made from 1 January to 31 December, in 2021 [25]. Additionally, information on the general characteristics of each type of medical institution was obtained from the HIRA. All the datasets used in this study were sourced from the Healthcare Bigdata Hub (https://opendata.hira.or.kr/home.do, accessed on 20 October 2022), an open system operated by the HIRA (Figure 1).

### 2.3. Outcome Variables and Independent Variables

The outcome variable of this study was the number of downloads from hospitals that introduced the m-PHR app. As described above, this variable is derived from the data obtained during an mPHR investigation conducted in May and June 2022. The numerical value of the number of downloads was sourced directly from the Google Play Store.

This study used three main independent variables of interest. They were originally numeric scales from one to five: first grade to fifth grade. The HIRA evaluated each hospital’s drug use and graded it using five scales. However, this study transformed them into a binary scale (1st grade or other) for ease of interpretation of the study results. The first variable was the results of a hospital evaluation of antibiotic drug use for acute upper respiratory infections. The second and third variables were hospital evaluation results on injection drug use and polypharmacy (multi-item prescription drugs).

The other independent variables representing the general characteristics of the hospitals chosen for this study were selected primarily from previous research. These variables included the hospital’s location, the years of operation, for-profit or public status, and affiliation with multi-hospital systems. Regarding location, most general hospitals in Korea are situated in urban areas. Therefore, Seoul and other mega-metropolitan cities were grouped together, while all other locations were classified as rural areas for our analysis. The operation period, measured in years since establishment, was defined as the duration of the hospital’s operation. For-profit status refers to private hospitals, including those run by medical foundations, whereas public hospitals encompass national or local government hospitals and public corporations. For multi-hospital systems, hospitals with the same name but different locations, or those under the same corporation but in different locations, were coded as having multiple hospitals. All the other hospitals were coded as not having multiple hospitals.

### 2.4. Statistical Analysis

To examine the relationship between the quality of medical services and the number of m-PHR app downloads, this study first analyzed the general characteristics of all hospitals with m-PHR applications. Specifically, it compared the number of m-PHR downloads between hospitals rated 1st and those rated lower using a *t*-test. For evaluating the number of drug items, the 1st group was limited to 16 hospitals, so a non-parametric test, the Wilcoxon–Mann–Whitney test, was conducted [26]. The correlations between the independent variables were assessed before the main analysis, and the variables with high correlations (e.g., number of hospital beds) were excluded from the analysis.

The outcome variable of this study is the number of downloads of m-PHR apps, which was not normally distributed, suggesting the use of a generalized linear model (GLM). Therefore, this study employed a GLM to examine the relationship between the number of m-PHR app downloads by hospitals and the independent variable of interest, while controlling for the general characteristics of the hospitals. Given that the dependent variable was not normally distributed, the distribution of the proposed model was diagnosed using the Modified Park Test, as suggested by Manning [27]. The results from this test were then applied to the GLM for the final analysis. Specifically, the Genmod procedure in the SAS program was used with the “Dist = Gamma” and “Link = Log” functions.

Statistical significance was set to *p* = 0.05. The statistical packages SAS/STAT ver. 9.4 (SAS Institute Inc., Cary, NC, USA) were used for statistical analysis [28].

## 3. Results

### 3.1. General Characteristics of the Study Subjects

For hospitals that had introduced the m-PHR system, this study investigated the relationship between the quality of care and the use of m-PHR applications, as measured by the number of downloads. The general characteristics of the hospitals are presented in Table 1. A total of 99 hospitals were analyzed. Approximately 56% of the participant hospitals were located in Seoul and other mega-metropolitan cities, and 75.8% were private hospitals. The average duration of operation since establishment was approximately 33 years. About 40% of the hospitals in this study were tertiary hospitals, and 78% were part of multi-hospital systems. The number of m-PHR downloads for hospitals with a 1st-grade evaluation in antibiotic prescription for acute upper respiratory diseases and polypharmacy was statistically significantly higher than that for hospitals without a 1st-grade evaluation (40,486 vs. 12,060 cases for antibiotic drug use; 76,312 vs. 24,675 cases for polypharmacy).

### 3.2. Relationship between the Quality of Care Evaluated by Antibiotic Drug Use for Upper Respiratory Diseases and the Number of m-PHR Application Downloads

Table 2 presents the relationship between the quality of care and m-PHR application downloads after controlling for hospital covariates. The results of the analysis indicate a close relationship between the two variables. The number of m-PHR downloads in hospitals with a good quality of care, as measured by antibiotic drug use, was 3.328 times higher than in hospitals without a 1st-grade rating, which was statistically significant (relative risk [RR], 3.328; 95% CI, 1.840 to 6.020; *p* < 0.001).

### 3.3. Relationship between the Quality of Care Evaluated by Injection Drug Use and the Number of m-PHR Application Downloads

Table 3 shows the association between the quality-of-care measures related to injection drug use and the m-PHR system downloads, after controlling for hospital covariates. Contrary to expectations, no statistically significant relationship was observed between the number of m-PHR system downloads and the quality of care as measured by injection drug use (RR, 1.426; 95% CI, 0.815 to 2.496; *p* = 0.214).

### 3.4. Relationship between the Quality of Care Evaluated by Polypharmacy and the Number of m-PHR Application Downloads

Table 4 presents the relationship between the quality of care, as measured by whether hospitals are rated 1st grade or not, with polypharmacy and the number of m-PHR application downloads. The number of m-PHR application downloads was statistically significantly higher in hospitals rated 1st grade compared to those that were not. The hospitals considered to provide high-quality care in terms of polypharmacy had 2.092 times more m-PHR application downloads than those not rated 1st grade for polypharmacy (RR, 2.092; 95% CI, 1.027 to 4.261; *p* = 0.042).

## 4. Discussion

This exploratory study aimed to investigate whether hospitals that provide high-quality care have more m-PHR app downloads than others. The background of this study is that the m-PHR app offers many functions related to patient convenience. Hospitals providing high-quality care are likely to prioritize patient convenience, including the use of m-PHR applications; thus, there may be a relationship between these two factors. This study measured the quality of care using three publicly disclosed hospital evaluation results: antibiotic drug use, injection drug use, and polypharmacy. This study found that the number of m-PHR app downloads was statistically significantly higher in hospitals providing high-quality care in antibiotic drug use and polypharmacy.

Regarding antibiotic drug use, this study found that the number of downloads of m-PHR apps was statistically higher in hospitals with high-quality care in terms of antibiotic drug use compared to other hospitals. As mentioned previously, m-PHR applications generally do not have any functions related to antibiotic drug use. Nevertheless, there was a statistically significant correlation between the number of m-PHR applications in hospitals and the quality of care regarding antibiotic drug use. The results of this study were similar to those of previous studies, which found that hospitals with active health information exchanges typically had good nursing manpower infrastructure, and hospitals providing high-quality care were more likely to adopt the m-PHR system [7,20]. This suggests that hospitals with excellent infrastructure and high-quality care might be interested in improving customer services, such as adopting and using the m-PHR system. Given that hospitals with sufficient medical staff and advanced medical equipment have introduced m-PHR applications and that PHR systems are effective in improving the quality of care for patients [13,29], it can be inferred that hospitals leading in providing high-quality care may be particularly interested in their patients’ use of m-PHR applications.

There was no correlation between the number of m-PHR app downloads and the quality of care measured by hospitals with a 1st-grade rating in injection drug evaluations. Generally, these findings make sense because m-PHR systems do not have any functions related to injection drug use. However, although this study did not find any statistically significant difference between the two factors, it is still interesting that hospitals with high-quality care for injection drugs had more m-PHR app downloads than those without a 1st-grade rating.

For quality of care measured by polypharmacy, defined as six or more drugs being prescribed by a hospital, those that received a 1st grade in polypharmacy had a higher number of m-PHR app downloads compared to the hospitals that did not. This analysis showed the same results as antibiotic drug use and is in direct contrast with the analysis results of injection drug use mentioned above. The interpretation of these results is similar to that of antibiotic drug use. Among the hospitals’ other independent variables, public foundation and the status of multi-hospital systems were positively associated with the number of m-PHR downloads.

Based on the results, hospitals that provide high-quality care in antibiotic use and polypharmacy are more likely to perform well in m-PHR, such as facilitating or assisting patients and their families in downloading m-PHR applications. It is highly probable that these hospitals had more m-PHR app downloads than others, although this may have been coincidental. Nonetheless, the observed relationship is noteworthy. Hospitals often receive favorable reviews from evaluations or consumers due to factors like staff kindness and facility cleanliness. In other words, hospitals with high ratings tend to excel in various areas, which might explain the observed relationship between these factors.

This study has several limitations. First, m-PHR applications do not provide information on antibiotic drug use, injection drug use, or polypharmacy. Therefore, it is reasonable to assume there is little direct relationship between these functions and m-PHR application downloads. This is the most critical limitation of this study. For this reason, this study was conducted as an exploratory one. Second, the number of m-PHR app downloads is likely to increase as the app provision period extends. However, these variables were not included in this study because they were not provided by the application. We expect that this limitation would minimally affect the study results, as the provision period would equally affect both groups. Lastly, the sample size of hospitals with a 1st grade in polypharmacy was small compared to that of the other groups (16 vs. 83 hospitals), which might have resulted in inflated test results due to a low statistical power.

This study is significant for the following three reasons. First, since all the data used were obtained from publicly available sources, the research results are objective and verifiable. Second, according to a recent survey on the status of domestic medical information systems conducted by the Korea Health and Medical Information Service, the m-PHR adoption rate in Korea was 61.9% for tertiary general hospitals and 22.6% for general hospitals with 300 or more beds [4]. Despite this, no studies have investigated the relationship between m-PHR app downloads and quality of care in Korea. Measurements must first take place in order to make some improvement [30]. This study anticipates that its results will significantly contribute to both the industrial and academic fields regarding m-PHR adoption and use and quality of care. Third, the study results are noteworthy because hospitals with high-quality care can be seen as leaders in other areas, such as the adoption and use of m-PHR applications.

## 5. Conclusions

This study examined the relationship between three quality-of-care indices—antibiotic drug use, injection drug use, and polypharmacy—and the number of m-PHR downloads in hospitals. The findings indicated that hospital leaders who provided high-quality care also played a significant role in enabling their patients and family members to download m-PHR applications. Although m-PHR applications did not have any functions directly related to drug use, hospitals with good overall evaluation results had a higher number of m-PHR downloads compared to those that did not. This was an exploratory study, and the results generally suggest that hospitals delivering high-quality care tend to perform well in other services, including m-PHR use. Various exploratory studies on m-PHR adoption and use are expected to be conducted in the near future.

## Figures and Tables

**Figure 1 healthcare-12-01064-f001:**
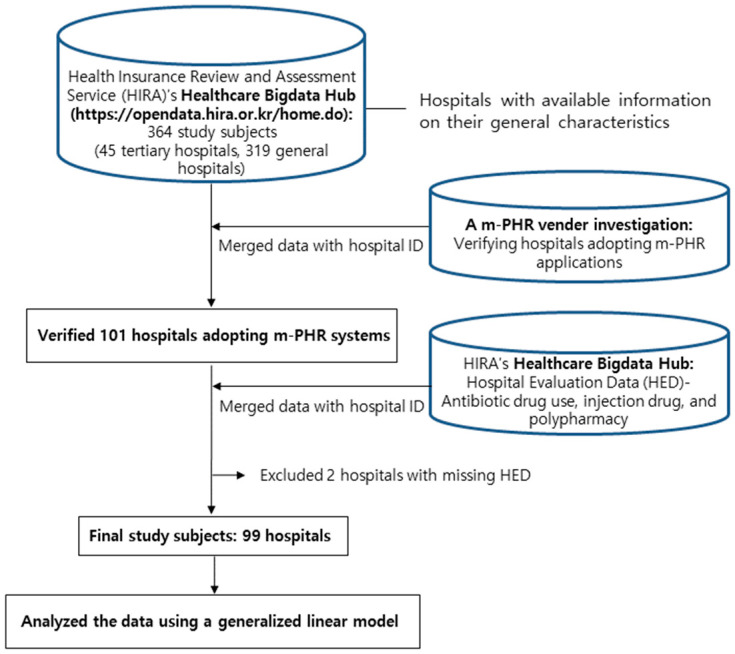
Flow of the data-selection process. m-PHR = mobile-based personal health record.

**Table 1 healthcare-12-01064-t001:** General characteristics of the study hospitals (N = 99).

Variables	% or Number	*p*-Value
All study subjects (hospitals) (%)	100.0	-
Location (%)		-
Seoul + Mega-metro cities	55.6	
The others	44.4	
For-profit status (%)		-
Private hospitals	75.8	
Public hospitals	24.2	
Years of operation (year)	33.2	-
Types of hospitals (%)		-
Tertiary (teaching) hospitals	40.4	
General hospitals	59.6	
Multi-hospital systems (%)		-
Yes	77.8	
No	22.2	
Anti-biotic drug prescription in upper respiratory disease		-
Number of m-PHR downloads of the 1st-grade group (N = 73)	40,486	0.003
Number of m-PHR downloads of the other group (N = 26)	12,060	
Prescription of injection drugs		
Number of m-PHR downloads of the 1st-grade group (N = 63)	36,269	0.419
Number of m-PHR downloads of the other group (N = 36)	27,335	
Polypharmacy drugs		
Number of m-PHR downloads of the 1st-grade group (N = 16)	76,312	0.002 ^1^
Number of m-PHR downloads of the other group (N = 83)	24,675	

^1^ *p*-value using the Wilcoxon–Mann–Whitney test.

**Table 2 healthcare-12-01064-t002:** Relationship between quality of care evaluated based on percent of antibiotic drug prescriptions and number of m-PHR application downloads.

Variables	Exp(β) ^1^	95% CI ^2^		*p*-Value
		LL	UL	
Foundation: For-profit (ref = public)	0.349	0.196	0.621	0.001
Location: Seoul + Mega-metro cities (ref = the others)	1.452	0.799	2.641	0.221
Tertiary hospital status (ref = general hospitals)	1.551	0.897	2.681	0.116
Years of operation (years)	1.000	0.980	1.021	0.984
Multi-hospital systems (ref = no)	1.897	0.948	3.795	0.070
Quality of care: 1st grade ^3^ (ref = the others)	3.328	1.840	6.020	<0.001

^1^ Exp(β): relative risk (RR), after controlling for all the variables in the table above; Exp: exponentiation; β, regression coefficient of GLM; ref: reference group; ^2^ CI, confidence interval; LL: lower limit; UL: upper limit; and ^3^ evaluated based on the percent of antibiotic drug prescriptions.

**Table 3 healthcare-12-01064-t003:** Relationship between the quality of care evaluated based on prescriptions of injection drugs and the number of m-PHR application downloads.

Variables	Exp(β) ^1^	95% CI ^2^		*p*-Value
		LL	UL	
Foundation: For-profit (ref = public)	0.339	0.183	0.627	0.001
Location: Seoul + Mega-metro cities (ref = the others)	1.229	0.676	2.233	0.499
Tertiary hospital status (ref = general hospitals)	1.510	0.842	2.710	0.167
Years of operation (years)	0.996	0.976	1.016	0.664
Multi-hospital systems (ref = no)	2.644	1.333	5.245	0.005
Quality of care: 1st grade ^3^ (ref = the others)	1.426	0.815	2.496	0.214

^1^ Exp(β): relative risk (RR) after controlling for all the variables in the table above; Exp: exponentiation; β, regression coefficient of GLM; ref: reference group; ^2^ CI, confidence interval; LL: lower limit; UL: upper limit; and ^3^ evaluated based on the percent of injection drugs.

**Table 4 healthcare-12-01064-t004:** Relationship between the quality of care measured by the number of drugs in prescription (1st grade vs. the others) and the number of m-PHR application downloads.

Variables	Exp(β) ^1^	95% CI ^2^		*p*-Value
		LL	UL	
Foundation: For-profit (ref = public)	0.426	0.229	0.790	0.007
Location: Seoul + Mega-metro cities (ref = the others)	1.141	0.642-	2.025	0.653
Tertiary hospital status (ref = general hospitals)	1.579	0.901	2.768	0.110
Years of operation (years)	0.999	0.980	1.018	0.899
Multi-hospital systems (ref = no)	2.476	1.257	4.879	0.009
Quality of care ^3^: 1st grade (ref = the others)	2.092	1.027	4.261	0.042

^1^ Exp(β): relative risk (RR), after controlling for all the variables in the table above; Exp: exponentiation; β, regression coefficient of GLM; ref: reference group; ^2^ CI, confidence interval; LL: lower limit; UL: upper limit; and ^3^ evaluated based on the percent of prescription drugs with polypharmacy.

## Data Availability

The data that support the findings of this study are available from the corresponding author upon reasonable request.

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
