# Peer review of "Quality of Care in Hospitals and the Use of Mobile-Based Personal Health Record Applications: An Exploratory Study Using National Hospital Evaluation Data"

_healthcare, 2024, doi:10.3390/healthcare12111064_

Round 1
Reviewer 1 Report
Comments and Suggestions for Authors
Good but need to reflect on my comments.
The paper is very interesting and insightful as it shed light on mobile-based personal health record applications. However, many points are unclearly clarified why m-PHR applications are important, leading to objectives, conducting method design, organizing finding structure, and conclusion.
1. Abstract
The introductory problem is uncleared, what is the purpose study, methods (setting, sample/sampling, procedures, data analysis) (see abstract section).
2. Introduction
The study lacks justification to conceptualize the m-PHR applications for the hospital. The manuscript uncleared theoretical gaps, conceptualize knowledge, objective, and questions.
Paragraph one uncleared
Paragraph two invalid clarifies the gaps
Paragraph three unclear how many factors are included in the study
Paragraph four unclear how to provide objectives
3. Literature
A wide and diverse literature has been used to develop the arguments reviewed in the paper. Given a limited number of studies on m-PHR applications for the hospital.
The review lack of deeper conceptualization theory, concept, and issues related to m-PHR applications. The literature lacks to have a theoretical support that will then lead into the identification of propositions or framework for m-PHR applications.
4. Methods
The methods are uncleared such as design, study setting, sampling procedures, measures, data collection, data analysis, and data rigor (see method section).
Moreover, the data collection is uncleared how many interview questions did the authors ask participants? What dimensions are reflected in the data?
Note: moved the IRB into the last page.
5. Results
The findings lack evidentiary support. Unorganized the main results, especially testing the statistics. The findings are lacking new practical contribution to m-PHR applications for the hospital.
6. Discussion
Should be clearly added the practical and organizational implications in the discussion section required.
Comments on the Quality of English LanguageThe paper may need the attention of a proofreader. The author(s) used pronouns. The quality of communication is poorly academic written. It is not only grammatical errors, but the writing style.
Author Response
Response to Reviewer 1 Comments
Point 1:
1. Abstract: The introductory problem is uncleared, what is the purpose study, methods (setting, sample/sampling, procedures, data analysis) (see abstract section).
Response 1: Thank you for your comments. We fully revised the abstract of the manuscript in order to reflect your comments focusing on the introductory problem, study objective, and methods. But we hope you understand that the abstract should be a total of about 200 words maximum according to the journal’s guideline.
Revised manuscript (Abstract): “The use of mobile-based personal health record (m-PHR) applications at the hospital level has been minimally studied. This study aimed to investigate the relationship between m-PHR use and quality of care. A cross-sectional study design was employed, analyzing data from 99 hospitals. Two data sources were utilized: a previous m-PHR investigation conducted from May 26 to June 30, 2022, and a hospital evaluation dataset on quality of care. The use of m-PHR applications was measured by the number of m-PHR application downloads. Three independent variables were assessed: quality of care in the use of antibiotic drugs, injection drugs, and polypharmacy with ≥6 drugs. A generalized linear model was used for analysis.” (p.1).
Point 2:
2. Introduction
The study lacks justification to conceptualize the m-PHR applications for the hospital. The manuscript uncleared theoretical gaps, conceptualize knowledge, objective, and questions.
Paragraph one uncleared
Paragraph two invalid clarifies the gaps
Paragraph three unclear how many factors are included in the study
Paragraph four unclear how to provide objectives
Response 2: Thank you for your comments. We revised our manuscript as much as we can in order to reflect your opinion. Please see below our point-by-point response to your comments.
Changed manuscript (Paragraph one): “Hospitals are places where a series of activities constantly take place to provide medical services. Most activities aim to improve the satisfaction of patients and their families, including the quality of care, and to increase the managerial efficiency of hos-pital operations. One of the recent trends is that many hospitals have introduced a mo-bile-based personal health record (m-PHR) system (hereinafter 'application,' 'app,' or 'system') into their practice settings [1-3]. Healthcare institutions, such as hospitals and clinics, provide patients with applications that allow them to view their visit records, schedules, medications, and test results [4,5].” (p.2).
Changed manuscript (Paragraph two): “These apps are useful tools for hospitals because they are convenient for patients. By using these apps, patients can see their schedules and the tests they have had in the past [6]. By accessing their records, patients can better understand their health and disease status and actively participate in treatment and prevention activities [7]. For these purposes, hospitals develop or deploy m-PHR apps that patients or their families can download from Google Play andtheApple Store.” (p.2).
Changed manuscript (Paragraph three): “A conspicuous phenomenon in the use of m-PHR applications is that the number of downloads varies depending on the hospital. Some hospitals have had 160,000 downloads since the app was launched, while others have significantly fewer [8]. Why do these variations among hospitals occur? If we can understand the causes of these differences, we can encourage more patients to download the applications. The more downloads, the higher the likelihood that patients will use them. To promote the use of m-PHR systems, we need to identify the factors influencing the number of m-PHR app downloads.” (p.2).
Changed manuscript (Paragraph four): “This study aimed to investigate the relationship between the quality of health care in hospitals and the use of m-PHRs as measured by the number of m-PHR application downloads.” (p.2).
Point 3:
3. Literature
A wide and diverse literature has been used to develop the arguments reviewed in the paper. Given a limited number of studies on m-PHR applications for the hospital.
The review lack of deeper conceptualization theory, concept, and issues related to m-PHR applications. The literature lacks to have a theoretical support that will then lead into the identification of propositions or framework for m-PHR applications.
Response 3: Thank you for your comments. We also agree with you. So, we added some manuscript text regarding organizational theory and empirical studeis supporting our argument. We also cited seveal references and, thus, all the reference numbers changed.
Added manuscript: “There have been no studies on the causes of this variation or related factors. This study argues that hospitals providing high-quality medical care will have more down-loads of m-PHR apps than others by assisting their patients in downloading these ap-plications. The Task-Technology Fit theory may support this argument. The theory postulates that individuals or organizations behave to maximize their performance by aligning their tasks with appropriate technology [9-11]. Here, “task” and "technology” could be defined or measured by a hospital's various activities related to improving the quality of care and "m-PHR application," respectively. This implies that m-PHR applications are well-suited for activities aimed at enhancing the quality of care, and hospitals interested in quality care are likely to adopt these applications in their opera-tions. Several empirical studies indirectly support this prediction. Hospitals providing good quality care are more likely to adopt various information technologies [12].” (p.2).
Point 4:
The methods are uncleared such as design, study setting, sampling procedures, measures, data collection, data analysis, and data rigor (see method section).
Moreover, the data collection is uncleared how many interview questions did the authors ask participants? What dimensions are reflected in the data?
Note: moved the IRB into the last page.
Response 4: Thank you for your comments. We revised the manuscript in order to reflect your comments. Please see below our point-by-point response to your comments
Changed manuscript (study design) (we corrected and changed our study design from case-control to a cross-sectional study design): “This study used a cross-sectional design, and the unit of analysis was hospitals with > 100 beds.” (p.2). “Regarding the research design, this study compared the number of downloads of m-PHR apps between hospitals that received a 1st grade rating in each of the three items used to evaluate the quality of care and hospitals that did not receive a 1st grade rating in Korea. The study categorized these evaluation results into 1st grade and non-1st grade institutions to facilitate easier interpretation for the reader.” (p.3)
Changed manuscript (study setting): “The study setting was Korea, which has a national health insurance(NHI) system. All citizens are free to choose a primary clinic-level medical institution and can then visit general hospitals or tertiary hospitals [22]. Because patients can freely choose general hospitals or tertiary hospitals, large hospitals make continuous efforts to retain their patients [23]. Since the m-PHR system offers many functions that provide various conveniences to patients [5, 14], it is very likely that hospitals with high-quality medical services will be interested in the introduction and use of m-PHR applications.” (p.3).
Manuscript location (Sampling procedures-this study used dataset that a previous study collected): “The first source comprised data from previous studies, which included information on the current status of m-PHR system implementation in hospitals and the number of m-PHR app downloads. The investigation of the m-PHR application in the previous study was conducted between May 26 and June 30, 2022. Using the hospital enrollment booklet, the study directly examined whether the hospital had a related m-PHR app through Google Play Store, Apple Play Store, Naver, and Google search engines. This research method was similar to those used in previous studies [13,24].” (p.3).
Manuscript location (measures of variables): “The outcome variable of this study was the number of downloads from hospitals that introduced the m-PHR app. As described above, this variable is derived from the data obtained during the mPHR investigation conducted in May and June 2022. The numerical value of the number of downloads was sourced directly from the Google Play Store.” (p.4). “This study used three main independent variables of interest. They were originally numeric scales from one to five: 1st grade to 5th grade. The HIRA evaluated each hos-pital’s drug use and graded it using five scales. However, this study transformed them into a binary scale (1st grade or other) for ease of interpretation of the study results. The first variable was the results of a hospital evaluation of antibiotic drug use for acute upper respiratory infections. The second and third variables were hospital evaluation results on injection drug use and polypharmacy (multi-item prescription drugs).” (p.4).
Manuscript location (data collection): “Two data sources were used in this study. The first source comprised data from previous studies, which included information on the current status of m-PHR system implementation in hospitals and the number of m-PHR app downloads.” (p.3) “The second source is public data from the HIRA, which includes data on the evaluation results of hospitals (hospital evaluation data) and their general characte-ristics. The HIRA evaluates hospitals using various indices. For this study, three items were selected as they can be comprehensively applied to all patients and are considered representative indicators for evaluating the quality of hospitals' medical services: the prescription rate of antibiotic use for acute upper respiratory tract diseases, the prescription rate of injection drugs, and polypharmacy evaluated based on the number of medicines prescribed. The HIRA assessed these items, announced the results with grades ranging from 1 to 5, and disclosed the results to the public.” (p.3).
Manuscript location (data analysis): “The outcome variable of this study is the number of downloads of m-PHR apps, which was not normally distributed, suggesting the use of a generalized linear model (GLM). Therefore, this study employed a GLM to examine the relationship between the number of m-PHR app downloads by hospitals and the independent variable of inter-est, while controlling for the general characteristics of the hospitals. Given that the de-pendent variable was not normally distributed, the distribution of the proposed model was diagnosed using the Modified Park Test, as suggested by Manning [27]. The results from this test were then applied to the GLM for the final analysis. Specifically, the Genmod procedure in the SAS program was used with the “Dist=Gamma” and “Link=Log” functions.” (p.5).
Changed manuscript (IRB): this study moved the manuscript on IRB to the end of the manuscript. (p.9).
Point 5:
5. Results
The findings lack evidentiary support. Unorganized the main results, especially testing the statistics. The findings are lacking new practical contribution to m-PHR applications for the hospital.
Response 5: Thank you for your opinion. We agree with you. Some readers may feel difficulty to understand the results of the Table 1. So, we fully reorganized Table 1. Regarding Table 2, 3, 4, UL and LL values given in Table 2.3 4 should be switched. So, we corrected them.
Reorganized manuscript: Table 1. (p.6)
Corrected manuscript: Table 2. (p.6), Table 3. (p.7), Table 4. (p.7),
Point 6:
6. Discussion
Should be clearly added the practical and organizational implications in the discussion section required.
Response 6: Thank you for your suggestion. We added some manuscript text to reflect your opinion. We think that the following manuscript is related with the practical and organizational implications.
Location of manuscript: “This study is significant for the following three reasons. First, since all data used were obtained from publicly available sources, the research results are objective and verifiable. Second, according to a recent survey on the status of domestic medical information systems conducted by the Korea Health and Medical Information Service, the m-PHR adoption rate in Korea was 61.9% for tertiary general hospitals and 22.6% for general hospitals with 300 or more beds [4]. Despite this, no studies have investigated the relationship between m-PHR app downloads and quality of care in Korea. Measurement must first take place in order to make some improvement [30]. This study anticipates that its results will significantly contribute to both the industrial and academic fields regarding m-PHR adoption and use, and quality of care. Third, the study results are noteworthy because hospitals with high-quality care can be seen as leaders in other areas, such as the adoption and use of m-PHR applications.” (p.9)
Point 7:
Comments on the Quality of English Language
The paper may need the attention of a proofreader. The author(s) used pronouns. The quality of communication is poorly academic written. It is not only grammatical errors, but the writing style.
Response 7: Thank you for your comment. We had had the proofread to our initial manuscript by a professional editing company before our submission. However, we had another proofread once again to this revised manuscript by a professional proofreading company following your comments. Thank you for your opinion.

Reviewer 2 Report
Comments and Suggestions for Authors
1) The N values given in Table 1 should be corrected to be small.
"Seoul+ Mega-metro cities (vs. the others)(%) 55.6 - For-profit hospitals: private(vs public) (%) 75.8" Other values must be written in the table for the percentage values to be valid.
2) UL and LL values given in Table 2.3 4 should be interchanged. The LL value must be given first.
3) Should the text given in the text be RR or OR? RR is a measure of risk generally obtained from cohort studies and intervention studies.
4) In the Study Design section of the study; "This study used a case-control design and the units of analysis were hospitals with > 100 beds." has been written.
5) There is not enough confidence in a complete case-control study. There is evidence that it is a cross-sectional study design rather than a case-control study.
6) If the cases are in 'general hospitals', are the controls in 'tertiary hospitals.'?
7) An explanation should be made regarding the relationship between the GLM model used and the logistic regression model. The table is generally reminiscent of the logistic regression model. It would be beneficial to add additional coefficients to the table in the specific GLM model used.
Author Response
Response to Reviewer 2 Comments
Point 1:
1) The N values given in Table 1 should be corrected to be small.
"Seoul+ Mega-metro cities (vs. the others)(%) 55.6 - For-profit hospitals: private(vs public) (%) 75.8" Other values must be written in the table for the percentage values to be valid.
Response 1: Thank you for your specific comments. We changed Table 1 in order to reflect your comment. In Table 1, N=99 is indicated in the title of Tabe1 and each variable is modified to have a % value.
Manuscript location: Table 1. (p.6)
Point 2:
2) UL and LL values given in Table 2.3 4 should be interchanged. The LL value must be given first.
Response 2: Thank you so much for your correction. You’re right. We corrected the manuscript. Thank you again!
Corrected manuscript location: Table 2, Table 3, and Table 4. (p.6-7)
Point 3:
3) Should the text given in the text be RR or OR? RR is a measure of risk generally obtained from cohort studies and intervention studies.
Response 3: Thank you for your question. Our outcome variable is the number of downloads of m-PHR apps which is numeric value. It was not normally distributed. So, we used GLM. In this case, interpretation of regression coefficients is relative risk (RR) such as % change of outcome variable per a one-unit change of an independent variable. We have a reference regarding this issue, but did not cite it. Please see the reference below.
Manuscript location: “Given that the dependent variable was not normally distributed, the distribution of the proposed model was diagnosed using the Modified Park Test, as suggested by Manning [27]. The results from this test were then applied to the GLM for the final analysis. Specifically, the Genmod procedure in the SAS program was used with the “Dist=Gamma” and “Link=Log” functions.” (p.6).
Reference regarding the RR (relative risk): Park YT, Lane C, Lee HJ, Lee J. Was size of healthcare institution a factor affecting changes in healthcare utilisation during the COVID-19 pandemic in Korea? A retrospective study design analysing national healthcare big data. BMJ Open. 2022 Dec 6;12(12): e064537. doi: 10.1136/bmjopen-2022-064537. PMID: 36600350; PMCID: PMC9729846.
Point 4:
4) In the Study Design section of the study; "This study used a case-control design and the units of analysis were hospitals with > 100 beds." has been written.
Response 4: Thank you for your comments. Yes, you’re right. So, we changed our manuscript to correct the manuscript.
Changed text: “This study used a cross-sectional design, and the unit of analysis was hospitals with > 100 beds.” (p.3).
Point 5:
5) There is not enough confidence in a complete case-control study. There is evidence that it is a cross-sectional study design rather than a case-control study.
Response 5: Thank you for your comments. Yes, you’re right. So, we changed our manuscript to reflect your comments.
Changed text (Abstract): “A cross-sectional study design was employed, analyzing data from 99 hospitals.” (p.1)
Changed text: “This study used a cross-sectional design, and the unit of analysis was hospitals with > 100 beds.” (p.3).
Changed text: “Regarding the research design, this study compared the number of downloads of m-PHR apps between hospitals that received a 1st grade rating in each of the three items used to evaluate the quality of care and hospitals that did not receive a 1st grade rating in Korea. The study categorized these evaluation results into 1st grade and non-1st grade institutions to facilitate easier interpretation for the reader.” (p.3).
Point 6:
6) If the cases are in 'general hospitals', are the controls in 'tertiary hospitals.'?
Response 6: Thank you for your question. If you ask about hospital types, then our answer is “yes”. Some of hospitals were general hospitals and the others were tertiary hospitals. We used a term, “ref” (reference group), in Table 2, 3, 4. (p.6-7).
Point 7:
7) An explanation should be made regarding the relationship between the GLM model used and the logistic regression model. The table is generally reminiscent of the logistic regression model. It would be beneficial to add additional coefficients to the table in the specific GLM model used.
Response 7: Thank you for your comments. Our main outcome variable is the number of downloads of m-PHR apps, but the number was not normally distributed. This suggests a generalized linear model (GLM). We added some manuscript text following your comments.
Added text: “The outcome variable of this study is the number of downloads of m-PHR apps, which was not normally distributed, suggesting the use of a generalized linear model (GLM). Therefore, this study employed a GLM to examine the relationship between the number of m-PHR app downloads by hospitals and the independent variable of inter-est, while controlling for the general characteristics of the hospitals.” (p.5).
Regarding additional coefficients, most of research articles used one regression coefficient column and thus we also followed them. But, we have added annotations to explain the regression coefficients of GLM to reflect your comment. Please see the manuscript text.
Manuscript location: “1Exp(β): relative risk (RR), after controlling all the variables in the table above, Exp: Exponentiation; β, regression coefficient of GLM (p.6, 7(Table 2, 3, 4)).

Round 2
Reviewer 1 Report
Comments and Suggestions for Authors
The revised version is well-suited and look good. All comments are revised and answered. I have accepted the manuscript. Good luck!
Comments on the Quality of English LanguageThe communication is good. However, some sentences are invalid meaning and grammatical errors.